# Evolutionary Analysis of OAT Gene Family in River and Swamp Buffalo: Potential Role of *SLCO3A1* Gene in Milk Performance

**DOI:** 10.3390/genes12091394

**Published:** 2021-09-10

**Authors:** Xiaoya Ma, Shasha Liang, Aixin Liang, Hossam E. Rushdi, Tingxian Deng

**Affiliations:** 1Key Laboratory of Buffalo Genetics, Breeding and Reproduction Technology, Buffalo Research Institute, Chinese Academy of Agricultural Sciences, Nanning 530001, China; maxiaoya8899@163.com (X.M.); cangshucangshu@126.com (S.L.); 2Key Laboratory of Agricultural Animal Genetics, Breeding and Reproduction of Ministry of Education, Huazhong Agricultural University, Wuhan 430070, China; lax.pipi@mail.hzau.edu.cn; 3Department of Animal Production, Faculty of Agriculture, Cairo University, 12613 Giza, Egypt; hosamrushdi@agr.cu.edu.eg

**Keywords:** buffalo, milk yield, organic anion transporter, gene family

## Abstract

The organic anion transporter (OAT) family is the subfamily of the solute carrier (SLC) superfamily, which plays a vital role in regulating essential nutrients in milk. However, little is known about the members’ identification, evolutionary basis, and function characteristics of OAT genes associated with milk performance in buffalo. Comparative genomic analyses were performed to identify the potential role of buffalo OAT genes in milk performance in this study. The results showed that a total of 10 and 7 OAT genes were identified in river buffalo and swamp buffalo, respectively. These sequences clustered into three groups based on their phylogenetic relationship and had similar motif patterns and gene structures in the same groups. Moreover, the river-specific expansions and homologous loss of OAT genes occurred in the two buffalo subspecies during the evolutionary process. Notably, the duplicated *SLCO3A1* gene specific to river buffalo showed higher expression level in mammary gland tissue than that of swamp buffalo. These findings highlight some promising candidate genes that could be potentially utilized to accelerate the genetic progress in buffalo breeding programs. However, the identified candidate genes require further validation in a larger cohort for use in the genomic selection of buffalo for milk production.

## 1. Introduction

The organic anion transporter (OAT) family is the secondary/tertiary active transporter protein falling into the subfamily of the solute carrier (SLC) superfamily [1]. This family, which has been determined to have six members, comprises a group of over 10 transmembrane proteins. Evidence revealed that the OAT family is present in many mammals and plays a vital role in regulating essential nutrients in milk [2]. Ibeagha-Awemu et al. [3] reported that the *SCLO3A1* gene was one of the candidate genes influencing cow milk traits. However, little is known about the function and characteristics of the OAT family affecting milk performance in buffalo.

Several similar genes constitute a particular gene family formed by duplication of an ancestral gene with similar biological functions. Identifying members, evolution, and expression analysis of the gene families has become the core content of comparative genome analysis, which helps in the systematic and comprehensive understanding of their putative functions. Evidence has revealed that gene duplication, including tandem and segmental duplication, plays a vital role in accelerating gene family expansion and genome evolution [4,5,6]. Moreover, gene duplication is considered as the single most important factor responsible for the formation of novel genes [7] and functional divergence [8]. Therefore, gene family analysis can help to better understand the evolution, function divergence, and formation history of the candidate genes. This strategy provides a useful reference for further investigating the biological mechanisms underlying species differentiation, biodiversity formation, and phenotypic diversity. In this regard, Rehman et al. [9] reported a gene family analysis of heat-shock proteins (HSP) that provides an understanding of HSPs in buffalo at the molecular level for the first time. In plants, Meng et al. [10] found that segmental duplication (SD) and whole-genome duplication (WGD) events are the major driving force for the expansion of the MADS-box gene family in pear, and their functional divergence of some orthologous genes originated from a common ancestor. The use of comparative genomics analysis will help understand the functional differentiation of the candidate gene and its effect on phenotypic diversity.

Water buffalo (*Bubalus bubalis*) is an important dairy animal mainly distributed in tropical and subtropical areas. Buffalo is divided into two subspecies: river buffalo (*Bu. bubalis bubalis*, 2*n* = 50) and swamp buffalo (*Bu. bubalis carabanesis*, 2*n* = 48), which differ in their chromosome number, morphology, and type of production specialization [11]. In general, river buffalo are mainly reared for milk production, whereas swamp buffalo are used for draught [12]. Compared to dairy cattle, lower milk performance is one of the main constraints in the current buffalo dairy industry. In this regard, we explored the evolution and function divergence of the OAT family genes in river and swamp buffalo based on the comparative genomic analysis. These results provide useful information for illuminating the biological functions of the OAT family genes affecting milk performance in buffalo.

## 2. Materials and Methods

### 2.1. Ethics Statements

Animal Care and Use Committee approval was not required for this study because the data were obtained from an existing database.

### 2.2. Identification of OAT Gene Family

A total of 7 mammals’ genome datasets were downloaded from the Genome database of NCBI and National Genomics Data Center (NGDC, BIG), namely, human (*GRCh38.p12*), cattle (*ARS-UCD1.2*), goat (*ARS1*), sheep (*Oar_rambouillet_v1.0*), horse (*EquCab3.0*), river buffalo (*UOA_WB_1*), and swamp buffalo (Accession number: GWHAAJZ00000000). The known protein sequences of the OAT gene family were downloaded from UniPort [13] and used to build hidden Markov model (HMM) profiles using HMMER v3.3.1 software [14]. All OAT proteins were searched in the species mentioned above by HMMER v3.3.1 software [14] with default parameters.

### 2.3. Gene Duplication Analysis of OAT Gene Family

Chromosomal locations of each OAT gene in river and swamp buffalo were obtained from their genome resources. The identification of orthologous OAT genes in river and swamp buffalo was analyzed and visualized by TBtools (v1.051) software [15] with the one-step MCScanX command. For each orthologous gene pair, the ratio between the nonsynonymous substitution rate (Ka) and the synonymous substitution rate (Ks) was computed by TBtools (v1.051) software [15]. The divergence time for each orthologous gene pair was estimated by T = Ks/2λ × 10^−6^ million years ago (Mya) [16], where T is the absolute time of divergence, Ks is the synonymous substitution rate, and λ is the clock-like rate in buffalo of 1.26 × 10^−8^ [17].

### 2.4. Sequence Analysis of OAT Gene Family

All identified OAT sequences were subjected to the ExPASy proteomics server to estimate their molecular weight (MW) and isoelectric points (pI). These protein sequences were further used for multiple alignments and the phylogeny tree construction implemented in MEGA-X software [18]. The motif pattern and gene structures of OAT family members were analyzed using the MEME platform and TBtools (v1.051) software [15], respectively. The motif element was annotated using the Pfam program. Moreover, two-kbp upstream promoter sequences from the start codon (ATG) of all orthologous OAT genes in river and swamp buffalo were also extracted by TBtools (v1.051) software [15]. They were further applied to analyze cis-elements using the TFBSTools (v3.11) package [19] in R.

### 2.5. Comparative Transcriptomic Analysis of the OAT Genes in Buffalo Subspecies

Two published RNA-seq data (BioProject: PRJEB4351 and BIGD: CRA002325) were selected and employed to explore the tissue expression patterns of the OAT gene family between the river and swamp buffalo. In the PRJEB4351 project, there are a total of 30 tissues from Mediterranean buffalo, ranging from lymphocytes to heart, bone marrow, spleen, and reproductive organs and representing a wide range of cell types. From the CRA002325 project, 24 tissues, including adipose, womb, different brain Brodmann areas, forelimb, hindlimb, medulla, heart, cerebellum, hypothalamus, pineal, kidney, tongue, spleen, ovary, liver, lung, and longissimus, from swamp buffalo were used in this study. In addition, RNA-seq data of buffalo milk samples at different stages of lactation (early, middle, and late) were downloaded from the NCBI SRA database with the accession number PRJNA453843, which was used to investigate the differential expression of the selected OAT genes in buffalo milk tissues. First, the quality control analysis was performed using TrimGalore v0.6.6 software for the raw data. Subsequently, the clean data were aligned to the buffalo genome (*UOA_WB_1*) using HISAT v2.2.1 software [20] with default parameters. The count matrix and transcripts per million (TPM) values for each gene were calculated using StringTie v2.1.3 software [21] and DESeq2 v3.11 [22] R-package, respectively. The one-to-one orthologous OAT genes in selected tissue samples were selected and compared between river and swamp buffalo, as described by Yu et al. [23]. The heat maps of TPM values for the buffalo OAT genes were performed using TBtools (v1.051) software [15].

## 3. Results

### 3.1. Identification and Phylogenetic Analysis of OAT Gene Family

To explore the potential function of the OAT family between river and swamp buffalo, we first performed family member identification. A total of 42 non-redundant protein sequences encoded by 17 OAT genes were predicted from the river and swamp buffalo genome (Table 1). Interestingly, these OAT genes clustered into three groups based on the phylogenetic analysis (Figure 1). Group 1 was composed of 5 OAT genes (*SLCO1C1*, *SLCO1A2*, *SCLO1B3*, *SCLO3A1*, and *SCLO2B1*), while Group 3 had only the *SCLO5A1* gene. Group 2 contained *SCLO4C1*, *SCLO4A1*, and *SCLO6A1* genes. The constructed dendrogram further showed that the buffalo OAT family was most closely related to the other representative mammals.

### 3.2. Chromosome Location and Gene Duplication Analysis of OAT Gene Family

A total of 10 OAT genes were found to be randomly distributed on 7 chromosomes of river buffalo, while the 7 OAT genes were randomly located on 4 chromosomes of swamp buffalo (Figure 2A). Interestingly, these genes were mainly located on the proximate or the distal ends of the chromosomes. Gene duplication analysis revealed that four pairs of OAT genes exhibited tandem duplication in the studied buffalo: *SLCO1B3*–*SLCO1A2* and *SLCO6A1*–*SLCO4C1* in river buffalo; *SLCO1C1*–*SLCO1A2* and *SLCO4C1*–*SLCO6A1* in swamp buffalo. Two pairs of genes (*SLCO3A1*–*SLCO4C1* and *SLCO4A1*–*SLCO5A1*) exhibited segmental duplication in river buffalo. Positive selection analyses further revealed that the Ka/Ks ratios of all the duplicated OAT gene pairs in buffalo were less than 1, suggesting that they underwent purifying selection. Divergence times of all duplicated gene pairs ranged from 41.545 to 96.155 Mya (Table 2). For the OAT gene family, five OAT gene pairs were orthologous between the two subspecies (Figure 2B), including *SLC2A1*, *SLCO3A1*–*SLCO4C1*, *SLCO4A1*, *SLCO5A1*, and *SLCO6A1*–*SLCO4C1* (Table 3). Divergence times of all orthologous gene pairs ranged from 0.063 to 55.210 Mya. More importantly, we found that the *SLCO3A1* and *SLCO4C1* gene pair was orthologous between the two subspecies (Figure 2B), while only the segmental duplication event occurred in the river buffalo population (Figure 2A). The divergence times of this orthologous gene pair occurred at 46.857 Mya (Table 3), while the segmental event started with 61.032 Mya (Table 2).

### 3.3. Sequence Characteristics of OAT Gene Family in Buffalo

To explore the structural characteristics of buffalo OAT members, motif pattern and gene structure analyses were performed considering their phylogenetic relationships (Appendix A). Nine and eight conserved motifs of OAT protein sequences in the river and swamp buffalo, respectively, were annotated as the OATP domain after the Pfam search (Appendix A), while motif1 was Kazal_2 dominant among them. Interestingly, the buffalo OAT gene family had a similar motif pattern in the same group. Although the introns and UTRs structure varied greatly between river (Appendix A) and swamp (Appendix A) buffalo, buffalo OAT genes in the same groups had similar exon and intron numbers, which confirm our previous classification process.

Moreover, we further explored the motif and TFBS patterns of the orthologous *SLCO3A1* and *SLCO4C1* gene pair between the two subspecies and the segmentally duplicated gene pair (*SLCO3A1*–*SLCO4C1*) in the river buffalo (Figure 3). The results revealed that the buffalo *SLCO4C1* had nine motifs, including six OATP domains, one OFCC1, and one Kazal_2 domain (Figure 3A), while the *SLCO3A1* had only four OATP domains. The TFBS analysis detected 19 *cis*-elements in the promoter regions of the two genes through the JASPAR2020 database (Figure 3A). Interestingly, the detected numbers of the homeodomain factors, fork head/winged helix factors, and C2H2 zinc finger factors were the top three members of the *SLCO4C1* gene in the river and swamp buffalo populations. For the *SLCO3A1* gene, the top three elements were the homeodomain factors, C2H2 zinc finger factors, and nuclear receptors with C4 zinc fingers (Figure 3B). These results suggested that the buffalo *SLCO3A1* gene might have different biological functions compared to the *SLCO4C1* gene.

### 3.4. Expression Pattern Analyses of OAT Gene Family in Buffalo

Using RNA-seq data from different tissues, we found that most OAT genes had a broad-spectrum expression pattern (Figure 4A,B). Meanwhile, different OAT genes in the same group had a similar expression pattern in the river (Figure 4A) and swamp (Figure 4B) buffalo. Among the clustering groups, Group 1 OAT genes (except for *SLCO1C1*) had a higher expression level than that of Group 2 (Figure 4A,B). Moreover, we compared the mRNA expression level of orthologous OAT genes between river and swamp buffalo, and the results showed that most genes (except for the *SLCO1A2*) in Group 1 had a relatively higher expression level in most river tissues, while the Group 2 genes with higher expression level were found in most swamp tissues (Figure 4C). Furthermore, four OAT genes were detected in the milk tissue of river buffalo (Figure 4D). Three OAT genes, namely, *SLCO2B1*, *SLCO3A1*, and *SLCO4A1*, exhibited higher mRNA expression levels than those of the *SLCO4C1* gene. Higher expression levels of *SLCO2B1* and *SLCO3A1* genes were observed at the early- and mid-lactation stages, while the greater expression level of *SLCO4A1* gene occurred in late lactation. These results suggest that different Group 1 OAT genes might have a stage-specific expression pattern.

## 4. Discussion

Gene family analysis has attracted widespread attention of both the research and farming communities owing to their contribution to a better understanding of the potential biological function affecting phenotypic diversity during evolution. It is well known that the OAT gene family plays a vital role in nutrient transport in the mammary gland [2,24]. In this context, our study is an attempt to systematically understand the OAT gene family’s potential functions affecting milk performance in buffalo through comparative genomics analysis. We identified 42 non-redundant protein sequences encoded by 17 OAT genes from the river and swamp buffalo genome. These protein sequences are clustered into three groups based on the phylogenetic analysis. The buffalo OAT gene family had a similar motif pattern and gene structure in the same buffalo group. To dissect the functional divergence of OAT gene family in the two buffalo subspecies, we performed tissue expression pattern analysis of the OAT genes. Our data showed that these OAT genes in the same group had a similar tissue expression pattern in river and swamp buffalo. Except for the *SLCO1C1* gene, Group 1 genes had higher expression levels than those of Group 2 genes. These results are in agreement with those obtained by Nigam et al. [1]. Evidence showed that genes in the Group 1 might be related to the uptake of hormones and organic ions in different tissues, such as the *SLCO1C1* gene for the uptake of thyroid hormones in brain tissues [25], *SLCO1A2* gene for the cellular uptake of organic ions in the liver [26], *SLCO2A1* gene for the uptake and clearance of prostaglandins in numerous tissues [27], and *SLCO2B1* gene for the regulation of placental uptake of sulfated steroids [28]. For Group 2, *SLCO4A1* gene has shown to be involved in transporting estrogen and its various precursors and metabolites, prostaglandin E2, and thyroid hormones [29,30]. The *SLCO4C1* gene was reported to be expressed at the basolateral membrane of the proximal tubule [31], and the *SCLO6A1* gene also predominantly expressed in the testis, with a limited expression in the epididymis [32]. Moreover, Group 3 has only the *SLCO5A1* gene, which might be a non-classical OAT family member involved in the processes of the cell shape reorganization [33].

Gene tandem and segmental duplication have significant roles in biological evolution. In this study, we observed that four gene pairs were duplicated in river buffalo through tandem or segmental duplication, while the swamp buffalo had two pairs of tandem duplicated genes. Collinearity analysis showed five orthologous genes in the river and swamp buffalo genomes. It was noted that the divergence times of the orthologous gene pair (*SLCO3A1*–*SLCO4C1*) between river and swamp buffalo occurred at 46.857 Mya. In addition, this gene pair had a segmental event in river buffalo, and its divergence time started with 61.032 Mya. These findings suggested that the *SLCO4C1* gene formed an orthologous gene (*SLCO3A1*) in river buffalo through a segmental duplication event. In contrast, the loss event of the *SLCO3A1* gene might have occurred in swamp buffalo during evolution. Moreover, RNA-data of mammary gland tissue in river buffalo showed that the Group 1 OAT genes, including *SLCO2B1*, *SLCO3A1*, and *SLCO4A1*, exhibited higher mRNA expression levels than that of *SLCO4C1* gene. Moreover, higher expression levels of *SLCO2B1* and *SLCO3A1* genes were observed at the early- and mid-lactation stages, while the higher expression level of *SLCO4A1* was noticed in late lactation. Both *SLCO3A1* and *SLCO4C1* genes were orthologous between the river and swamp buffalo. More importantly, we found that the segmental duplication event might be involved in the expression pattern of the *SLCO4C1* gene in the river buffalo, resulting in a novel river-specific duplicated gene (*SLCO3A1*). On the other hand, the swamp buffalo lost this duplicated gene. Previous studies reported that segmental duplication events play a vital role in forming novel gene functions [7,34,35]. In this regard, we presume that the segmental duplication event of OAT gene family has a positive effect on enhancing milk production performance in river buffalo. In other words, the loss of *SLCO3A1* gene in swamp buffalo may be one of the reasons leading to the low milk performance of the swamp buffalo compared to that of river buffalo. However, this inference needs to be confirmed through further investigation.

Considering the function characteristics of *SLCO3A1* and *SLCO4C1* genes during evolution, we further explored the TFBS patterns of the two orthologous genes between the two subspecies. The results showed that the detected numbers of the Homeo-domain factors, Fork-head/winged helix factors, and C2H2 zinc finger factors were the top three members of the *SLCO4C1* gene in river and swamp buffalo populations. For the *SLCO3A1* gene, the top three elements were the Homeo-domain factors, C2H2 zinc finger factors, and Nuclear receptors with C4 zinc fingers (Figure 3B). Evidence revealed that Homeo-domain factors play a fundamental role in a diverse set of functions including body plan specification, pattern formation, and cell fate determination during metazoan development [36]. The Fork-head/winged helix factors were reported to play important regulatory roles in a wide variety of biological functions, including cell proliferation, immunity, apoptosis, and metabolism [37]. The C2H2 zinc finger factors were known to be involved in many cellular processes including DNA-binding, RNA-binding, protein-protein interaction, and protein folding, etc. [38]. Porter et al. [39] reported that the Nuclear receptors with C4 zinc fingers have a unique role in the maintenance of cellular homeostasis, gene expression regulation in embryogenesis, and tissue development. These results suggested that the buffalo *SLCO3A1* gene might have different biological functions compared to the *SLCO4C1* gene.

## 5. Conclusions

In summary, we detected 17 buffalo OAT genes. The river-specific expansions and homologous loss of OAT genes were observed in the two buffalo subspecies during the evolutionary process. Notably, the duplicated *SLCO3A1* gene of river buffalo showed a higher expression level in the mammary gland tissue in comparison with that of the swamp buffalo. These results provide useful information for illuminating the biological functions of the OAT family genes affecting milk performance in buffalo.

## Figures and Tables

**Figure 1 genes-12-01394-f001:**
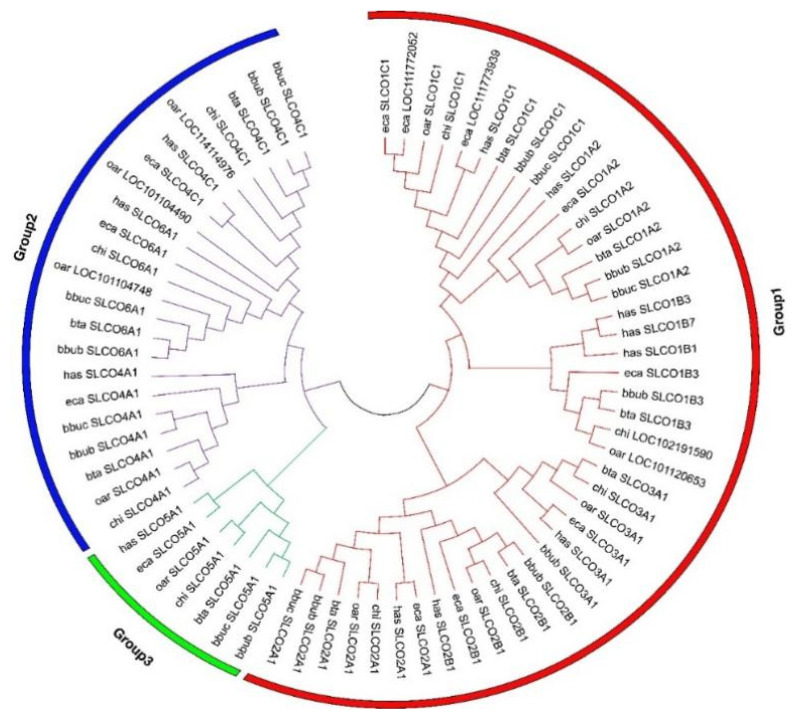
Phylogenetic relationship of OAT proteins in seven mammals. Line with different colors indicates different groups. Different colored circles indicate different groups. River buffalo: bbub; swamp buffalo: bbuc; cattle: bta; goat: chi; sheep: oar; horse: eca; human: has.

**Figure 2 genes-12-01394-f002:**
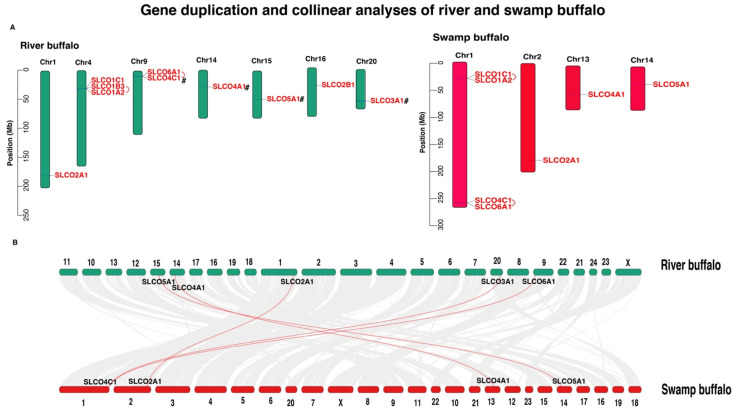
Gene duplication of river and swamp buffalo OAT genes (**A**) and their collinear analysis (**B**)**.** The tandem duplicated genes are marked by a red line, and segmentally duplicated genes are indicated by “#.”

**Figure 3 genes-12-01394-f003:**
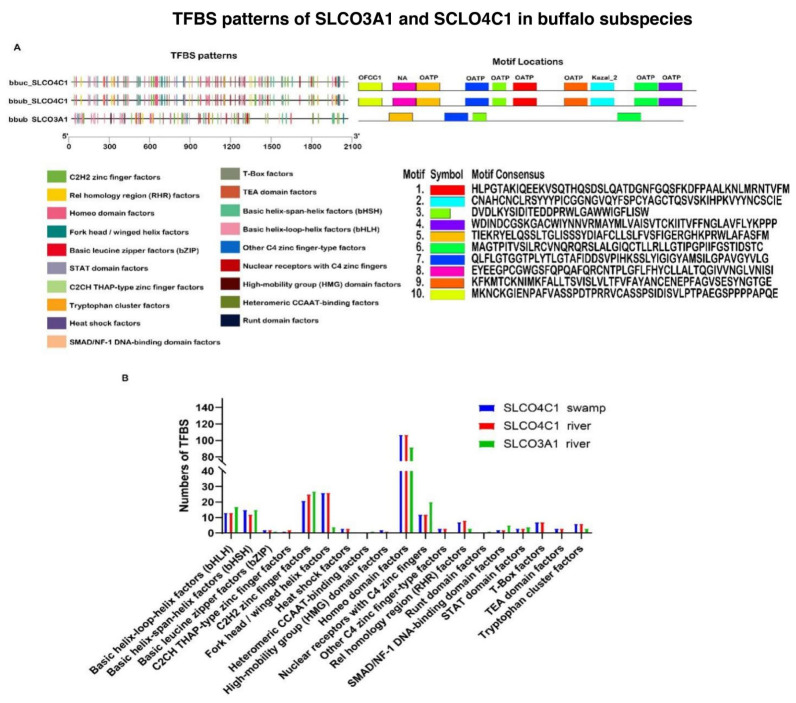
The gene structure and TFBS patterns of SLCO3A1 and SCLO4C1 in buffalo subspecies. (**A**) The TFBS and motif patterns of SLCO3A1 and SCLO4C1 between the two subspecies. (**B**) The number of TFBS of SLCO3A1 and SCLO4C1 between the two subspecies.

**Figure 4 genes-12-01394-f004:**
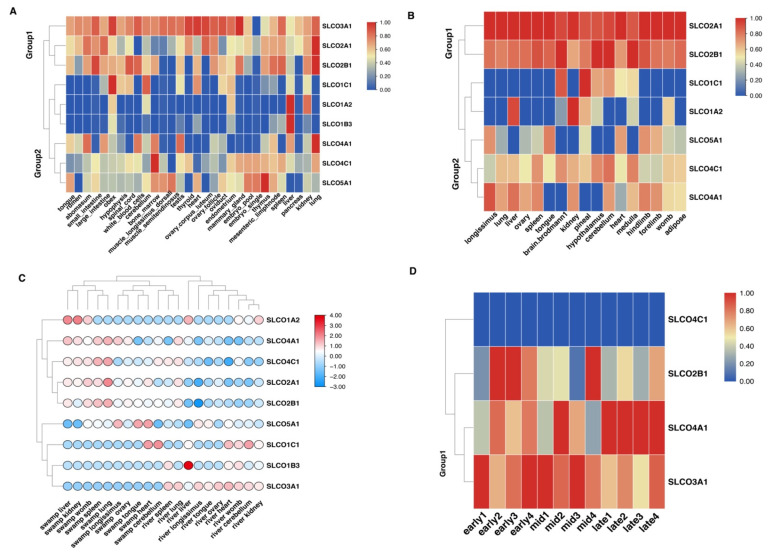
The expression patterns of OAT genes in buffalo subspecies. (**A**) The expression distribution of OAT genes in river buffalo tissues. (**B**) The expression distribution of OAT genes in swamp buffalo tissues. (**C**) The expression differential patterns of orthologous OAT genes between river and swamp buffalo. (**D**) The expression patterns of OAT genes in buffalo milk samples at different lactation.

**Table 1 genes-12-01394-t001:** Features of the predicted OAT protein sequences in buffalo.

Species	Gene	CHR	Protein ID	Length	Mw (kDa)	pI
river	*SLCO1A2*	Chr4	XP_006067688.1	666	73.79	7.09
river	*SLCO1C1*	Chr4	XP_006067684.2	712	78.46	9.02
river	*SLCO1B3*	Chr4	XP_025139143.1	689	75.45	8.71
river	*SLCO3A1*	Chr20	XP_025127060.1	692	74.38	6.94
river	*SLCO2A1*	Chr1	XP_006051482.1	644	70.2	8.81
river	*SLCO2B1*	Chr16	XP_006074131.1	708	76.67	7.93
river	*SLCO4C1*	Chr9	XP_006069687.1	719	77.4	7.00
river	*SLCO5A1*	Chr15	XP_006073179.2	846	92.13	7.89
river	*SLCO4A1*	Chr14	XP_006072365.2	761	81.19	7.55
river	*SLCO6A1*	Chr9	XP_025147965.1	731	79.8	8.52
swamp	*SLCO1C1*	Chr1	GWHPAAJZ000564	1222	133.66	8.31
swamp	*SLCO1A2*	Chr1	GWHPAAJZ000565	666	73.81	7.09
swamp	*SLCO2A1*	Chr2	GWHPAAJZ009028	644	70.2	8.81
swamp	*SLCO4C1*	Chr1	GWHPAAJZ002472	719	77.49	6.82
swamp	*SLCO5A1*	Chr14	GWHPAAJZ005023	846	92.13	7.89
swamp	*SLCO4A1*	Chr13	GWHPAAJZ004480	761	81.2	7.64
swamp	*SLCO6A1*	Chr1	GWHPAAJZ002473	933	100.78	7.97

Note. CHR: buffalo chromosome; Mw: molecular weight; pI: isoelectric points.

**Table 2 genes-12-01394-t002:** The Ka/Ks ratios and divergence times for each pair of duplicated OAT genes in buffalo subspecies.

Subspecies	Seq1	Seq2	Type	Ka	Ks	Ka/Ks	Date (Mya)
River	*SLCO1B3*	*SLCO1A2*	tandem	0.573	1.664	0.344	66.014
	*SLCO6A1*	*SLCO4C1*	tandem	0.546	1.047	0.522	41.545
	*SLCO3A1*	*SLCO4C1*	Segment	0.746	1.538	0.485	61.032
	*SLCO4A1*	*SLCO5A1*	Segment	0.718	2.180	0.329	86.518
Swamp	*SLCO1C1*	*SLCO1A2*	tandem	0.512	2.423	0.211	96.155
	*SLCO4C1*	*SLCO6A1*	tandem	0.564	1.063	0.530	42.198

Note. Ka: nonsynonymous substitution rate; Ks: synonymous substitution rate; Mya: million years ago.

**Table 3 genes-12-01394-t003:** The Ka/Ks ratios and divergence times for each pair of orthologous OAT genes in buffalo.

Seq1/River Buffalo	Seq2/Swamp Buffalo	Ka	Ks	Ka/Ks	Date (Mya)
*SLCO2A1*	*SLCO2A1*	0.001	0.008	0.118	0.302
*SLCO6A1*	*SLCO4C1*	0.861	1.391	0.619	55.210
*SLCO4A1*	*SLCO4A1*	0.723	1.165	0.620	46.238
*SLCO5A1*	*SLCO5A1*	0.000	0.002	0.000	0.063
*SLCO3A1*	*SLCO4C1*	0.811	1.181	0.686	46.857

Note. Ka: nonsynonymous substitution rate; Ks: synonymous substitution rate; Mya: million years ago.

## Data Availability

Not applicable.

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
