# Peer review of "Evolutionary Analysis of OAT Gene Family in River and Swamp Buffalo: Potential Role of SLCO3A1 Gene in Milk Performance"

_genes, 2021, doi:10.3390/genes12091394_

Round 1

Reviewer 1 Report

This is an interesting article, which main goal is to investigate OAT genes and their association with milk performance in buffalo. I think that the manuscript does not need many changes, but it can be improved by applying some alterations as listed below.

The paper does not provide a clearly and literally precise aim of the study. One can read the article and understand what authors wanted to do, but it is best to write the aim and hypothesis used in the study so that it would be more clear whether the hypothesis was proven or not. Each research should be done because it wants to prove or dismiss the hypothesis.

In Figures 2 – 4 it is really hard (or even not possible) to read the content. Font is to small or is disfigured. I would suggest changing fonts to be bigger or changing the size of the whole figure. Also if it is not possible to add bigger figures in supplement.

Tables – each table should be self-explanatory, one should not be expected to go back to the text and search for explanation for the abbreviations. It is easier if all abbreviations are explained either in the table title or the footnote e.g. what CHR, pI, KA, KS, Date(Mya) means.

Conclusion – would it be possible to give a conclusion that would explain what is the more utilitarian aspect of the obtained result.  “illuminating the biological functions” is a broad expression.

Author Response

Reviewer1

  1. The paper does not provide a clearly and literally precise aim of the study. One can read the article and understand what authors wanted to do, but it is best to write the aim and hypothesis used in the study so that it would be more clear whether the hypothesis was proven or not. Each research should be done because it wants to prove or dismiss the hypothesis.

Au: Thank you for comment. This study aims to conduct the members identification, evolutionary basis, and function characteristics of OAT family associated with milk performance in buffalo. This information has added in the abstract section, please check it at line 14-17.

  1. In Figures 2 – 4 it is really hard (or even not possible) to read the content. Font is to small or is disfigured. I would suggest changing fonts to be bigger or changing the size of the whole figure. Also if it is not possible to add bigger figures in supplement.

Au: Thank you for comment. We have followed your suggestion. Some figures have updated, which maybe make readers understand the current results more clearly.

  1. Tables – each table should be self-explanatory, one should not be expected to go back to the text and search for explanation for the abbreviations. It is easier if all abbreviations are explained either in the table title or the footnote e.g. what CHR, pI, KA, KS, Date(Mya) means.

Au: Thank you for comment, we have fixed them and explained the abbreviations in Tables. please check them.

  1. Conclusion – would it be possible to give a conclusion that would explain what is the more utilitarian aspect of the obtained result. “illuminating the biological functions” is a broad expression.

Au: Thank you for comment. In the present study, we found the river-specific duplicated SLCO3A1 gene showed a higher expression level in mammary gland tissue than that of swamp buffalo. In contrast, the SLCO3A1 gene was not found in the swamp buffalo. It means that the segmental duplication event of OAT gene family might have a positive effect on enhancing milk production performance in river buffalo. In other words, the loss of SLCO3A1 gene in swamp buffalo may be one of the reasons leading to the low milk performance of the swamp buffalo compared to that of river buffalo. However, this inference needs to be confirmed through further investigation. In addition, our current data did not reveal the specific role of this gene on milk performance in buffalo. Thus, we just use a broad expression of the obtained result to state our idea. Undeniably, our current results provide useful information to further explore the specific biological functions of this gene on buffalo milk performance.

Reviewer 2 Report

The manuscript: “Evolutionary analysis of OAT gene family in river and swamp buffalo: potential role of SLCO3A1 gene in milk performance” presents some in-silico analyses to link the Organic Anion Transporter (OAT) gene family to the mink performance in buffalo. Although the research has merit, the current investigations are purely based on bioinformatics approaches and publically available data. Therefore,  the results are not novel enough to publish in the Genes journal as a full paper.

In particular, in section 2.5, the authors did not use qRT-PCR but using the data from RNAseq. From lines 116-120, it is not clear how the authors perform the DE analyses.

In Figure 4, the authors indicated the expression of the genes, but the changes in the gene expression are not statistically tests. Therefore, the implications of these results might not be correct.  

The quality of Figure 3 is very poor. The letters in Motif consensus are not readable.

It is not also clear which color scale is used in Figure 4.

Minors:

 Line 88: 10-6 to 10-6

Table 1; Please explain some abbreviations in columns: CHR, Mw, pI

Author Response

Reviewer2

  1. The manuscript: “Evolutionary analysis of OAT gene family in river and swamp buffalo: potential role of SLCO3A1 gene in milk performance” presents some in-silico analyses to link the Organic Anion Transporter (OAT) gene family to the mink performance in buffalo. Although the research has merit, the current investigations are purely based on bioinformatics approaches and publically available data. Therefore, the results are not novel enough to publish in the Genes journal as a full paper.

Au: Although the current investigations are purely based on bioinformatics approaches and publicly available data, we performed the members identification, gene duplication and express analysis of OAT family in buffalo. Notably, we found some interesting things: the river-specific expansions and homologous loss of OAT genes occurred between the two buffalo subspecies during the evolutionary process. Interestingly, the river-specific duplicated SLCO3A1 gene showed a higher expression level in mammary gland tissue than that of swamp buffalo. In other words, the loss of SLCO3A1 gene in swamp buffalo may be one of the reasons leading to the low milk performance of the swamp buffalo compared to that of river buffalo. However, this inference needs to be confirmed through further investigation. Thus, we think the current results were suitable for the requirements of Gene journal.

  1. In particular, in section 2.5, the authors did not use qRT-PCR but using the data from RNAseq. From lines 116-120, it is not clear how the authors perform the DE analyses.

Au: Thank you for comment. Yes, here only RNA-seq was used to explore the expression of OAT family genes. We have deleted the word of qRT-PCR. Additional information has been displayed in the revised manuscript. Please check it. Moreover, the DE analyses were not used in the present study. So, the information on the DE analysis was not provided.

  1. In Figure 4, the authors indicated the expression of the genes, but the changes in the gene expression are not statistically tests. Therefore, the implications of these results might not be correct.

Au: Thank you for comment. According to your suggestion, we reanalyzed the RNA-seq Data. We also performed the comparative analysis of OAT family genes in some tissues between river and swamp buffalo. Briefly, the one-to-one orthologous OAT genes in selected tissue samples were compared between river and swamp buffalo, as described by Yu et al. (Yu, L.; Wang, G.D.; Ruan, J.; et al. Genomic analysis of snub-nosed monkeys (Rhinopithecus) identifies genes and processes related to high-altitude adaptation. Nat Genet 2016, 48, 947-952, doi:10.1038/ng.3615.). Please check the new results in the main text.

  1. The quality of Figure 3 is very poor. The letters in Motif consensus are not readable.

Au: Thank you for comment, we have updated this Figure. Please check it.

  1. It is not also clear which color scale is used in Figure 4.

Au: Thank you for comment, we have updated this Figure. Please check it.

  1. Line 88: 10-6 to 10-6

Au: Thank you for comment, we have fixed it.

  1. Table 1; Please explain some abbreviations in columns: CHR, Mw, pI

Au: Thank you for comment, we have fixed them and explained the abbreviations in Table 1.

Round 2

Reviewer 2 Report

Thank you for the response. All my comments have been addressed. I suggest the authors improve the quality of the figure 2. The figure 2 and 3 should be have the title, before explaining A and B.
Table 2 and 3 ; The authors could remove the word analysis or change it to the results of analysis of ...
In table 2, is this species or sub-species?
In Figure 3, there are several figure in the 3.A, please add the legend to explain them. 
Figure 4: What did the authors mean milk tissues?

Author Response

Reviewer2

  1. I suggest the authors improve the quality of the figure 2. The figure 2 and 3 should be have the title, before explaining A and B.

Au: Thank you for comment. The titles have been added in the two Figures. Please check them.
2.Table 2 and 3 ; The authors could remove the word analysis or change it to the results of analysis of ...

Au: Thank you for comment. We have fixed them, please check them.
3. In table 2, is this species or sub-species?

Au: Thank you for comment. It is a subspecies. We have add this information to the table.
4. In Figure 3, there are several figure in the 3.A, please add the legend to explain them.

Au: Thank you for comment. We have fixed them, please check them.
5. Figure 4: What did the authors mean milk tissues?

Au: Thank you for comment. The RNA-seq data used in this study from the buffalo milk at different lactation. So, milk samples were maybe more suitable than the milk tissues. We have used the milk samples instead of the milk tissues in the revised version.